# The Role of Extracellular Proteases and Extracellular Matrix Remodeling in the Pre-Metastatic Niche

**DOI:** 10.3390/biom15121696

**Published:** 2025-12-05

**Authors:** Gillian C. Okura, Alamelu G. Bharadwaj, David M. Waisman

**Affiliations:** 1Department of Pathology, Dalhousie University, Halifax, NS B3H 1X5, Canada; gillian.okura@dal.ca (G.C.O.); alamelu.bharadwaj@dal.ca (A.G.B.); 2Department of Pathology and Biochemistry and Molecular Biology, Dalhousie University, Halifax, NS B3H 1X5, Canada

**Keywords:** pre-metastatic niche, matrix metalloproteases, serine proteases, plasminogen, extracellular matrix remodeling, cathepsins, myeloid-derived suppressor cells, S100A10, fibroblasts, neutrophils, uPA, uPAR, plasmin

## Abstract

The premetastatic niche (PMN) represents a specialized microenvironment established in distant organs before the arrival of metastatic cells. This concept has fundamentally altered our understanding of cancer progression, shifting it from a random event-driven process to an orchestrated one. This review examines the critical role of extracellular proteases in PMN formation, focusing on matrix metalloproteinases (MMPs), serine proteases, and cysteine cathepsins that collectively orchestrate extracellular matrix remodeling, immune modulation, and vascular permeability changes essential for metastatic colonization. Key findings demonstrate that MMP9 and MMP2 facilitate basement membrane degradation and the recruitment of bone marrow-derived cells. At the same time, tissue inhibitor of metalloproteinase-1 (TIMP-1) promotes organ-specific hepatic PMN recruitment through neutrophil recruitment mechanisms. The plasminogen–plasmin system emerges as a master regulator through its broad-spectrum proteolytic activity and ability to activate downstream proteases, with S100A10-mediated plasmin generation providing mechanistic pathways for remote PMN conditioning. Neutrophil elastase and cathepsin G contribute to the degradation of anti-angiogenic proteins, thereby creating pro-metastatic microenvironments. These protease-mediated mechanisms represent the earliest interventional window in metastatic progression, offering therapeutic potential to prevent niche formation rather than treat established metastases. However, significant methodological challenges remain, including the need for organ-specific biomarkers, improved in vivo methods for measuring protease activity, and a better understanding of temporal PMN dynamics across different target organs.

## 1. Introduction

Tumor metastasis is the primary cause of mortality in many cancer types; hence, a synergized focus on understanding the mechanisms driving cancer metastasis is crucial to developing novel therapeutic strategies. Our understanding of the role of cancer cells and the primary tumor microenvironment in promoting cancer metastasis has significantly improved in the past decades. Emerging studies suggest that the host microenvironment not only contributes to metastatic dissemination but also to colonization of secondary organs and overall survival. These secondary sites, established before the clinically detectable metastases, are termed pre-metastatic niches (PMNs). In this review, we will discuss the role of proteases in PMN at secondary sites and evaluate therapeutic options for targeting these PMNs.

Stephen Paget first postulated the “seed” and “soil’ hypothesis in 1889, establishing that primary tumors systematically prepare distant organs for their arrival and colonization [1]. He describes the incoming cancer cells as the “seed” and the distant organs as the “soil”. This hypothesis was further strengthened by Isiah Fidler, who suggested that metastatic colonization occurs only at specific organs [2].

The PMN is a specialized microenvironment in distant organs that is modified in response to tumor-secreted factors, including soluble factors and extracellular vesicles (EVs) (reviewed in [3]). These factors prepare the microenvironment to be receptive to tumor cells and support their colonization of these distant organs. Metastasis-driving genes were first proposed in 2003 by Kang et al., who demonstrated that genes regulating vascular permeability, angiogenesis, extracellular matrix (ECM) remodeling, bone marrow-derived cell (BMDC) recruitment, and immunosuppression were all upregulated in cell line subpopulations with elevated metastatic activity [4]. This was the first proposal that genes could regulate metastasis to a specific organ, highlighting processes that are now recognized as hallmarks of PMN formation.

The term PMN was first coined by Kaplan et al. in 2005, in groundbreaking studies that showed the importance of Vascular Endothelial Growth Factor Receptor 1 (VEGFR1)- expressing BMDCs in forming pre-metastatic sites in the lungs before the arrival of tumor cells [5].

The identification of the PMN represents a paradigm shift in our understanding of the metastatic process and has enhanced our understanding of metastasis progression. From being known as a random set of spontaneous events, it is now defined as a carefully orchestrated sequence of events. The PMN provides favorable conditions for tumor colonization in an otherwise hostile host environment, thereby facilitating the proliferation and survival of these cells.

Proteases were initially identified and characterized for their functional roles in the general degradation of proteins. Still, over the past few decades, growing evidence has strengthened their role in cancer cell invasion, metastasis, and modulation of the tumor microenvironment. There are four classes of proteases, namely, matrix metalloproteinases (MMPs), serine proteases, cysteine cathepsins, and aspartic proteases. These proteases are classified based on their mechanism of enzymatic reactions and catalysis, and they encode more than 3% of the human [6,7]. Proteases play a vital role in normal physiological processes, whereas their dysregulation of proteolytic activity and function causes various pathological conditions, including cancer. One of the several critical functions of proteases in cancer progression involves the degradation of the ECM, thereby enhancing the invasion of cancer cells into the neighboring and distant tissues. The ECM surrounds the tumor microenvironment and is primarily composed of a complex network of non-cellular structural proteins, including collagen, glycoproteins, proteoglycans, laminin, and fibronectin [8,9,10]. It not only provides structural support for the various cellular components of the tumor microenvironment but also serves as a reservoir for growth factors and cytokines, which promote tumor progression and metastasis [11].

Several excellent reviews have discussed the role of proteases in cancer progression [7,12,13,14]. The focus of this review will be to highlight recent developments that focus on the roles and types of proteases involved in PMN formation.

## 2. PMN Formation

The process of PMN formation comprises four distinct phases: *priming*, *licensing*, *recruitment*, and *colonization*. Tumor-derived secreted factors (TDSFs), resident and recruited BMDCs, and the local stromal microenvironment are key to PMN formation and eventual metastasis (Figure 1). The PMN establishment involves four steps: alterations in vascular permeability, activation of stromal cells, remodeling of the ECM, and infiltration of immunosuppressive immune populations to allow the colonization of circulating tumor cells [15,16]. These components play roles through the ordered, stepwise progression by which distant organs are prepared and rendered hospitable for disseminated tumor cells (DTCs).

ECM remodeling at the pre-metastatic site/organ is necessary not only to support the survival and growth of incoming tumor cells but also to enhance the recruitment and survival of BMDCs, which further support tumor cell survival and growth. The various ECM components, such as fibronectin, collagen, and laminin, provide a niche environment that directs the survival, adhesion, and colonization of circulating tumor cells [17].

The formation of the PMN begins when hypoxic conditions in the rapidly growing tumor trigger the release of TDSFs and tumor-derived extracellular vesicles (TDEVs). Under hypoxic conditions within the primary tumor, HIF-1α in tumor cells is upregulated, translocates to the nucleus, and activates transcription of target genes, including the VEGF gene promoter, leading to increased VEGF expression [18]. When tumor cells release VEGF, it binds to VEGFR2 receptors on endothelial cells [19], triggering receptor autophosphorylation and activation of downstream signaling pathways. This process elevates intracellular calcium concentrations [20,21], which simulates the activity of endothelial nitric oxide synthase (eNOS) [22]. Activated eNOS generates NO and prostacyclin, both of which relax endothelial cell–cell junctions by modifying vascular endothelial cadherin (VE-Cadherin) complexes [23]. VEGF further amplifies vascular permeability by inducing platelet-activating factor (PAF) synthesis, creating a feedforward loop that enables TDSFs to enter the systemic circulation [24].

HIF-1α activation further drives the secretion of VEGF and transforming growth factor-β (TGF-β) into the circulation, which stimulates fibroblasts and stromal cells in the secondary site to produce lysyl oxidase (LOX), an amine oxidase critical for collagen cross-linking [25]. This enzymatic activity modifies the ECM architecture at distant sites by oxidizing lysine residues in collagen IV, thereby creating a stiffened scaffold [25]. Simultaneously, VEGF and TGF-β induce phenotypic switching in perivascular cells, suppressing traditional smooth muscle markers while upregulating the pluripotency factor *KLF4*, which promotes a dedifferentiated state conducive to fibronectin deposition [5]. This fibronectin deposition increases the rigidity and promotes the adhesion of cells recruited to the secondary site [26,27]. LOX can induce fibronectin production in resident fibroblasts, thereby stiffening the matrix and promoting the accumulation of matrix remodeling proteins, such as MMP9 [28,29]. This coordination degrades the basement membrane, exposing cryptic fibronectin sites that establish a permissive scaffold for DTCs. Fibroblast activation further amplifies fibronectin production, establishing a provisional matrix that serves as a docking site for integrin-expressing myeloid cells and DTCs [5,30,31].

The immunomodulatory functions of TDSFs are equally critical for PMN priming. VEGF synergizes with TGF-β and tumor necrosis factor-α (TNF-α) to stimulate resident myeloid cells in the primitive PMN, triggering the production of proinflammatory cytokines S100A8 and S100A9 [32,33]. These alarmins create a chemotactic gradient for CD11b^+^ myeloid-derived suppressor cells (MDSCs) and macrophage recruitment by triggering the assembly of cytoskeletal contractile elements, facilitating their movement into the niche [34]. Recruited macrophages secrete MMP9 to degrade basement membranes further and enhance vascular permeability [35]. Macrophages can activate acute inflammatory responses in the microenvironment by synthesizing acute-phase proteins, including serum amyloid A (SAA), C-reactive protein, and fibrinogen, to sustain inflammation [34]. Secreted SAA can recruit Gr1^+^CD11b^+^ myeloid cells that inhibit cytotoxic CD8^+^ T cell activity [33,35]. Concurrently, LOX-generated ECM stiffness activates pro-fibrotic signalling (e.g., FAK/Src) in resident fibroblasts, fostering an immune-excluded microenvironment [25,36,37,38]. CXCL13, secreted from the primary tumor, induces resident cells to reprogram and produce pro-thrombotic extracellular vesicles (EVs) [39]. This cascade establishes an initial permissive microenvironment characterized by ECM stiffening, immune evasion, and angiogenic priming.

EVs have emerged as critical initiators in this process, particularly during the earliest priming stage. EVs encompass diverse membranous structures classified primarily by size and composition. Small EVs include exosomes (30–150 nm) and exomeres (non-membranous nanovesicles), while large EVs (>150 nm) comprise ectosomes, microvesicles, blebbisomes, and apoptotic bodies [40,41]. These membranous-enclosed or non-membranous particles contain varied biological cargo that enables sophisticated intercellular communication by encapsulating signalling molecules, protecting them from degradation [42,43,44,45] and immune detection [46,47,48,49]. Exosome-mediated intracellular communication is primarily mediated by membrane proteins binding to target cells, directly fusing with the target cell membrane or binding to target cell membrane proteins to activate signal pathways in the target cell [41,50,51].

Tumor hypoxia significantly influences the production and content of EVs during the priming stage. Cancer cells produce substantially more TDEVs compared to normoxic conditions, and these TDEVs carry altered cargo with enhanced metastasis-promoting potential [52,53,54,55]. Under hypoxic conditions, exosomes are upregulated in proteins involved in glycolysis, the coagulation system, and innate immune response pathways [56]. HIF-1α enhances TDEVs secretion and enriches their cargo with pro-metastatic molecules such as matrix remodeling (e.g., LOX, plasminogen), metabolic reprogramming (e.g., phosphoglycerate kinase-1), and proliferation (e.g., MET proto-oncogene) [51,57,58]. TDEVs prime distant organs by increasing vascular endothelial permeability via the transfer of regulatory genes, such as *uPAR*, *VEGF*, and TNF-α [58,59]. TDEVs can alter the expression of junction proteins (e.g., VE-cadherin) [60,61,62], thereby further stimulating angiogenesis to support the recruitment of BMDCs.

Hypoxia can alter the miRNA content of cancer-derived EVs, with significant implications for locomotion, cell migration, cell motility, vascular development, and metabolic processes, potentially enhancing their reprogramming to a distant site [63]. For example, hypoxic lung cancer cells secrete exosomes that carry HIF-1α-induced miR-494, which promotes angiogenesis by suppressing PTEN and activating the Akt/eNOS pathway in endothelial cells [64]. Similarly, hypoxic leukemia cells release exosomes that enhance tube formation in endothelial cells through miR-210 [65]. These TDEVs amplify the priming signals necessary for effective PMN establishment.

A defining feature of TDEVs is their ability to direct organ-specific metastasis through distinct molecular signatures. Specific integrin patterns on the EV membrane, such as α6β1, α6β4, αvβ5, and Gβ3, can direct their uptake by resident cells in the PMN [31,66]. This selective targeting enables tissue-specific EV accumulation. Upon uptake by organ-specific cells, these integrins activate Src phosphorylation and pro-inflammatory *S100A8* and *S100A4* gene expression, initiating structural changes in distant organs [31]. TDEVs also participate in fundamental ECM modifications that stiffen the matrix, promote fibronectin deposition, and recruit BMDCs [67,68]. Tumor-derived exosome LOX is present on the surface of exosomes and can contribute to ECM stiffening through collagen crosslinking [69]. TDEVs can carry MMPs as cargo, enabling the transfer of these active enzymes between cells to facilitate ECM remodeling [70,71,72,73,74]. Resident cells can take up signal-carrying exosomes and release TGF-β, thereby promoting further fibronectin deposition [67,68].

Tumor-derived exosomes activate the toll-like receptor 2 (TLR2)/NF-κβ signaling axis in macrophages, driving HIF-1α-mediated upregulation of GLUT-1. This glycolytic reprogramming enhances glucose uptake and shifts the metabolism away from traditional oxidative phosphorylation [68]. These metabolic shifts promote macrophage polarization toward an immunosuppressive phenotype through NF-κβ-dependent programmed death ligand 1 (PD-L1) expression and lactate overproduction. This establishes an immune-evasive microenvironment critical for PMN formation during the priming phase [65,72]. The selective delivery of bioactive molecules to specific distant organs transforms these sites into receptive microenvironments for DTCs, preparing them for BMDC recruitment.

## 3. Cellular Determinants of the PMN

The formation of PMN depends on fibroblasts, neutrophils, and other cell types, which direct the intricate cellular and molecular changes that establish a welcoming environment for metastatic cells before they arrive. The formation of PMN by cancer-associated fibroblasts (CAFs) occurs through multiple mechanisms, including exosome regulation, metabolic changes, ECM modification, and the production of immunosuppressive factors [75]. The expression of COX-2 by lung-resident fibroblasts leads to the production of prostaglandins such as prostaglandin E2 (PGE2), which disrupts the function of dendritic cells and suppressive monocytes, creating an environment that supports breast cancer metastasis. Lung-resident stromal fibroblasts serve as master regulators of PMN formation and ECM remodeling, responding to tumor-derived signals by producing both ECM proteins (fibronectin, collagen types I-V) and chemokines that facilitate the recruitment of immune cells [75,76,77,78,79]. The PMN contains neutrophils, which function as essential components that modify the microenvironment by switching between N1 and N2 phenotypes with opposing effects on tumor growth. The process of PMN recruitment leads neutrophils to change their behavior by releasing S100A8, S100A9, and Bv8 molecules and forming neutrophil extracellular traps (NETs) that trap tumor cells via integrin β1 binding [80].

Multiple cell types beyond fibroblasts and neutrophils play essential roles in the establishment of PMN structures. The PMN contains myeloid-derived suppressor cells (MDSCs), which include monocytic (M-MDSC) and polymorphonuclear (PMN-MDSC) subtypes. These suppress immune function through arginase-1 activity, nitric oxide production, and lipid transfer mechanisms, which block dendritic cell antigen presentation [81,82]. The presence of Tregs in PMNs results from tumor-exosome-mediated upregulation of CCL1 in fibroblasts, which then attracts Tregs through CCL1-CCR8 signaling. The activation of the Aryl Hydrocarbon Receptor (AHR) by GM-CSF in macrophages leads to their conversion into immunosuppressive cells, which form PMNs by expressing PD-L1 and differentiating into Treg cells. In fact, the process of PMN formation begins when VEGFR1+ hematopoietic progenitor cells from bone marrow establish cellular clusters at future metastatic sites before tumor cells reach the area [5]. Endothelial cells regulate PMN sites by controlling vascular permeability and blood vessel formation, as well as by suppressing the immune system. VEGF stimulates PGE2 production and the migration of BMDCs. The PMN suppresses natural killer (NK) cells through PGE2, TGF-β, and reactive oxygen species (ROS) signals from neutrophils, while B cell numbers decrease during micrometastatic development. The early formation of metastatic niches depends on platelets, which quickly cluster around tumor cells, producing CXCL5 and CXCL7 chemokines that attract granulocytes via CXCR2 receptors, thereby creating essential but short-lived metastatic sites shortly after tumor cell arrest. The various cellular determinants function synergistically through protease-mediated ECM remodeling to alter the normal microenvironment at metastatic sites and organs into tumor-permissive metastatic niches [17,81,83,84].

## 4. Proteases Involved in PMN Formation

In cancer progression, altered proteolysis, induced by both the host tissue and the tumor microenvironment, disrupts tissue homeostasis. Deregulated proteolysis leads to uncontrolled tumor growth, tissue remodeling, inflammation, cancer cell invasion, ultimately leading to metastasis [85]. In the context of the PMN role, proteases are more complex and layered, as both cancer-cell-secreted proteases and local proteases activated at the PMN site can contribute to tumor cell colonization and homing to the sites. However, recent studies have systematically elucidated the contributions of proteases in the tumor secretome and of local organ-specific proteases to PMN function, using sophisticated technologies. These studies collectively demonstrate the advantages of local protease activation versus tumor-secreted proteases in the PMN development. Local proteases can be rapidly activated and inactivated in response to local microenvironmental cues.

Furthermore, local proteases can be directly expressed by recruited tissue-resident cells, whereas tumor-secreted proteases arrive via the circulation and hence require specific protective mechanisms. The local expression and activation of proteases are mediated by several resident and recruited cell types, including tissue-resident macrophages, recruited neutrophils, endothelial cells, and BMDCs, rather than relying on circulating tumor-derived proteases. Local and tumor-secreted proteases modulate one or more steps in PMN development at secondary sites, including reprogramming the ECM, altering vascular permeability, and recruiting immunosuppressive cells. In the following sections, we will focus on proteases and their specific roles in PMN development. The various proteases in PMN have been summarized inTable 1.

### 4.1. Matrix Metalloproteases (MMPs)

Matrix metalloproteinases (MMPs) are a family of 28 calcium-dependent, zinc-containing endopeptidases belonging to the metzincin superfamily that collectively degrade all components of the extracellular matrix and regulate numerous biological processes beyond simple proteolysis [94]. MMPs represent the most extensively studied protease family in PMN formation [95,96,97,98]. MMPs serve as both direct mediators of ECM remodeling and activators of growth factors and cytokines. MMP9, in particular, has been extensively studied. The proteolytic activities driven by MMPs collectively establish a tumor-conducive primary microenvironment, facilitating tumor progression, angiogenesis, epithelial–mesenchymal transition, and the early dissemination of cancer cells from the primary tumor [98]. Not only that, MMPs play several fundamental roles in preparing distant organs, such as the lungs, liver, and brain, for metastatic colonization of cancer cells across multiple cancer types. These functions include enhancing vascular permeability and the extravasation of circulating cancer cells into the PMN, liberating growth factors, and, most importantly, recruiting immune cells to create an immunosuppressive environment within the PMN. Interestingly, MMPs are expressed in an organ-specific manner, with the liver expressing and inducing high endogenous MMP activity. In contrast, the lung PMN requires cancer-cell-derived MMP expression, including MMP2 and MMP9 [99].

Two important studies have identified and elucidated the critical role of MMP9 in the PMN lung to promote metastasis. Hiratsuka et al. showed that MMP-9 is specifically induced in pre-metastatic lung endothelial cells and macrophages by distant primary tumors via VEGFR-1/Flt-1 tyrosine kinase signaling, establishing a critical mechanism for lung-specific metastasis [35]. The key study by Kaplan et al. established that VEGFR1+ hematopoietic progenitor cells from the bone marrow (not the tumor) home to pre-metastatic sites and locally express MMP-9. The important finding was that tumor-derived growth factors upregulate fibronectin expression by the lung-resident fibroblasts. These growth factors also recruit VEGFR1+ hematopoietic cells, which, once recruited to the lung PMN, produce MMP9 locally to degrade the local basement membrane and secrete Kit-ligand and VEGFA, further accelerating the extravasation of more VEGFR1+ cells into the lung niche. This cascade of events precedes the arrival of tumor cells in the permissive PMN sites [5]. Similarly, studies by Huang et al. demonstrated that MMP-3 and MMP-10, along with angiopoietin-2, are upregulated in the lung by primary tumor cells and function synergistically to disrupt vascular integrity and facilitate lung metastasis by creating a tumor cell-permissive lung microenvironment [86]. In another landmark study, Erler et al. observed that LOX is the key determinant of lung PMN in mouse breast cancer models. They showed that hypoxic breast cancer cells secrete LOX, which, upon circulation, cross-links Collagen IV in the lung basement membrane. This further results in the recruitment of Cd11b+ myeloid cells. These cells express and activate MMP2, which degrades Collagen IV, thereby further recruiting BMDCs and tumor cells to the lungs [25]. A pre-metastatic conditioning study using exosomes derived from hypoxic prostate cancer cells revealed the highest MMP (MMP9 and MMP2) activity in prostate cancer target metastatic organs, such as the liver, kidney, and spleen [56]. These studies demonstrate the importance of host-induced systemic protease activity in PMN development and the critical roles of various MMPs in preparing PMNs. Recent studies suggest that neutrophils are the key cellular determinants of lung PMN establishment [100]. Single-cell transcriptomics profiling of lung tissue in the transgenic PyMT-MMTV mouse model revealed increased infiltration of N2-type neutrophils, which expressed increased levels of angiogenic and inflammatory markers. Of particular interest to this review, increased MMP9 expression was associated with N2-type neutrophils. The proteolytic activity of MMP2 and MMP14 contributes to collagen IV degradation at PMN sites, generating collagen peptides that function as chemoattractants, amplifying myeloid cell recruitment to the PMN [101]. Additionally, MMPs such as MMP10 and MMP13 increase vascular permeability during PMN formation, which facilitates the extravasation of circulating tumor cells [101,102,103].

In contrast, tissue inhibitor of metalloproteinases (TIMP)-1 is critical for liver metastasis in colorectal cancer models. TIMP-1 is a small molecular weight secreted protein initially identified as an endogenous inhibitor of MMPs [104]. It has been shown to play dual and contradictory roles in cancer progression [105], where elevated TIMP1 is consistently associated with poor prognosis in multiple cancer types [105,106,107,108]. Specifically in colorectal cancer, TIMP-1 plays a critical role in liver PMN formation through stromal-derived factor-1 (CXCL12)/CXCR4-dependent neutrophil recruitment [88]. In pancreatic cancer models, TIMP-1 promotes liver PMN by binding to CD63 on hepatic stellate cells, inducing a positive feedback loop that leads to increased TIMP-1 expression and enhanced neutrophil recruitment to the PMN via CXCL12 [109]. In summary, the organ-specific proteolytic landscape fundamentally determines TIMP-1′s role in PMN formation. While the liver’s constitutively high gelatinolytic activity creates a permissive environment where TIMP-1 functions as a pro-metastatic signaling molecule rather than a protease inhibitor, the lung’s requirement for induced MMP expression renders TIMP-1 dispensable for pre-metastatic conditioning in pulmonary tissues [88].

Exosome-derived ADAM17 from colorectal cancer (CRC) is a critical mediator of vascular barrier disruption and promotes PMN formation in the liver in CRC. This study demonstrated, using mouse models, that high-ADAM17-expressing exosomes increased vascular leakage, DTC counts, and VE-cadherin expression at the tumor’s invasive front, leading to more frequent and larger liver and lung metastases. Specifically, depleting ADAM17 expression in exosomes and using ADAM17 inhibitors restored VE-cadherin expression, vascular permeability, and decreased metastatic burden. Furthermore, this study showed that circulating exosomal ADAM17 levels were significantly higher in patients with metastatic CRC than in those without metastasis and healthy controls. Elevated exosomal ADAM17 correlated with poor prognosis and higher DTC counts, suggesting its potential as a blood-based biomarker for predicting CRC metastasis [91].

### 4.2. Serine Proteases

Serine proteases are a category of enzymes that cleave peptide bonds in proteins using a serine residue in their active sites [110]. They employ a catalytic triad consisting of serine, histidine, and aspartate to hydrolyze peptide bonds with high specificity and efficiency [111]. They are further classified into several types based on the amino acid at the cleavage site in the protein/peptide to be hydrolyzed. These include trypsin-like (examples: thrombin, plasmin, complement system proteases) [112], chymotrypsin-like (examples: elastin, cathepsin G, granzymes) [113,114,115], subtilisin-like proteases (examples: furin, PC1/PC3, PACE4) [116], membrane-anchored serine proteases (examples: TMPRSS2, TMPRSS4, matriptase, hepsin) [117,118], and plasminogen activation system (tissue plasminogen activator, urokinase plasminogen activator, plasminogen-activator-inhibitors) [119,120,121,122]. Serine proteases play essential physiological roles in digestion, coagulation, fibrinolysis, immune responses, and the maintenance of tissue homeostasis. However, dysregulated expression and activity of these proteases drive cancer progression and metastatic dissemination. The serine protease superfamily comprises a diversity of enzymes; only a restricted subset of enzymes has been identified to play a role in PMN biology and development, including neutrophil-derived proteases (elastase, cathepsin G) and some of the components of the plasminogen activation system, while the roles of other serine protease family members in this context remain largely unexplored. We will explore the most prevalent serine proteases and their roles in PMN development.

Among the various serine proteases, the strongest experimental evidence for their role in PMNs exists only for Neutrophil Elastase (NE) and Cathepsin G (CG), particularly in the context of inflammation-induced metastatic outgrowth in the lungs [87]. The authors showed that Lipopolysaccharide (LPS)-induced inflammation recruited neutrophils to the lungs. Furthermore, neutrophil-derived NE and CG degraded the anti-angiogenic protein thrombospondin-1 (TSP-1). Using genetically engineered NE^−/−^/CG^−/−^ double-knockout mice and bone marrow reconstitution studies, the researchers demonstrated that eliminating these specific serine proteases preserved TSP-1 integrity and significantly reduced metastatic colonization efficiency in a B16 mouse melanoma model. While this study provides compelling evidence for the roles of NE and CG in priming the PMN before tumor cell arrival in the lungs, a critical limitation is the use of artificial inflammation and experimental metastasis models rather than orthotopic tumor models. This circumvents the natural progression of primary tumor-driven PMN formation. Nevertheless, it will be interesting to evaluate if NE and CG are elevated in the PMN in other primary-tumor- or secretome-conditioned lung PMN models.

The plasminogen–plasmin system represents another crucial protease pathway in PMN development, as tumor cells utilize plasminogen receptor–plasminogen activator complexes to generate plasmin at specific times during PMN formation and in particular locations within the PMN [123,124]. Plasminogen is an inactive zymogen secreted by the liver. It is converted to a broad-spectrum serine protease plasmin by plasminogen activators such as urokinase plasminogen activator (uPA) and tissue plasminogen activator (tPA) [123]. The activity of tPA and uPA are regulated by the presence of plasminogen-activator-inhibitors 1 and 2 (PAI-1 and PAI-2) [125]. The presence of plasminogen receptors on cancer and stromal cells accelerates the slow rate of conversion of plasminogen to plasmin by the activators. Thus, plasminogen receptors are key regulators of plasmin generation. Several Pg receptors have been identified and characterized to accelerate plasmin generation [126,127,128]. Among the myriad of Pg receptors, Histone H2B, HMGB1, alpha-enolase, plasminogen-Rkt, cytokeratin 8, and S100A10 (p11) have been extensively characterized for their role as Pg receptors in cancer progression [123].

Plasmin orchestrates degradation of fibrin clots (fibrinolysis-clot lysis), inflammation, wound healing, and cancer metastasis. Specifically, plasmin mediates metastasis by directly remodeling the ECM proteins and indirectly by activating MMP9. It also plays a crucial role in cell signaling [129,130]. As a broad-spectrum serine protease, plasmin contributes directly or indirectly to the regulation of inflammation [131]. It is well established that the plasmin-plasminogen system plays a vital role in remodeling the ECM at the primary tumor site to allow the invasive escape of cancer cells [123]. Plasmin-mediated degradation of extracellular matrix proteins, particularly fibronectin, enhances tumor cell motility through increased α5β1 integrin-mediated interactions. Plasmin also activates several MMP zymogens, including MMP2 and MMP9. Furthermore, plasmin cleaves cell surface receptors, such as protease-activated receptors (PARs), generating functionally important cleavage products that activate outside-in signaling pathways critical for tumor progression [130]. However, a significant gap remains in understanding how plasminogen and plasmin modulate the pre-metastatic and metastatic niche. These could be attributed to a lack of accurate methodologies for determining in vivo plasmin activity. Moreover, organ-specific analysis of the proteins involved is complicated by their systemic circulation. However, it is compelling to hypothesize that plasmin is the key regulator of PMN function due to its roles in activating MMPs, regulating cell signaling, and mobilizing bone marrow-derived cells. Future studies should systematically elucidate the role of this protease in PMN development using plasminogen-KO mouse models and/or plasmin inhibitors, with spatio-temporal analysis across various PMN organs, such as the brain, lungs, and liver.

S100A10 (p11), complexed with annexin A2 to form the AIIt heterotetramer, serves as an important cellular plasminogen receptor [132,133,134,135]. This complex binds plasminogen via its carboxyl-terminal lysine residues, facilitating efficient plasmin generation while protecting plasmin from inactivation by α2-antiplasmin. Beyond its canonical role in plasmin regulation, S100A10 also regulates the expression of cell-surface receptors and ion channels, contributing to its pleiotropic effects in cancer [136,137,138].

Our laboratory and others have established that S100A10 plays an oncogenic role in multiple cancer types [134,139,140,141,142,143,144,145,146,147,148,149,150]. S100A10 overexpression, which occurs in many cancers, including breast cancers, enhances tumor cell invasion, migration, and metastasis through increased ECM degradation. In the MMTV-PyMT transgenic mammary tumor model, loss of S100A10 resulted in a significant decrease in spontaneous pulmonary metastatic burden [139]. Notably, Li et al. recently demonstrated that cancer cell secretomes increase S100A10 expression in lung fibroblasts, leading to the expression of CXCL1 and CXCL8 chemokines and the recruitment of MDSCs, which are essential for PMN formation [92]. S100A10-deficient mice demonstrated significant protection from lung metastasis, and pharmacological inhibition of S100A10 substantially reduced the metastatic burden. However, these investigators did not examine whether S100A10’s role in PMN formation occurs through plasmin-dependent or independent mechanisms.

Several lines of evidence have shown that tumor-derived EVs and exosomes serve as sophisticated delivery vehicles for multiple components of the plasminogen activation system. For instance, uPA [151,152] and plasminogen activator inhibitor-1 (PAI-1) are detected in tumor-derived EVs. Key plasminogen receptors, including annexin A2 [93], S100A10 [92,153], alpha-enolase (ENO1) [154], are detected in tumor-derived vesicles, and histone H2B in tumor-derived EVs [155] for systemic transport to distant organs. These EV-encapsulated components represent a novel mechanism by which primary tumors can remotely condition future metastatic sites, with uPA-containing exosomes demonstrated to enhance metastatic behavior both locally and at distant sites in osteosarcoma models [151]. At the same time, exosomal annexin A2 promotes PMN formation in lung and brain tissues through its function as a tPA/plasminogen co-receptor [93]. The clinical significance of this EV-mediated delivery of the plasminogen system is underscored by the identification of uPAR+ extracellular vesicles as robust biomarkers for metastatic disease [156] and by the correlation of exosomal PAI-1 levels with tumor progression and immune evasion [157,158]. However, despite the compelling presence of these plasminogen activation components in tumor-derived EVs and their established roles in metastatic progression, there remains a critical knowledge gap regarding the precise molecular mechanisms by which EV-delivered uPA, tPA, plasminogen receptors, and regulatory factors coordinate to activate plasmin specifically within PMN environments, highlighting the need for future research to definitively establish the functional plasminogen activation cascades that occur when tumor-derived EVs encounter distant organ microenvironments during niche preparation.

Understanding the function of the plasmin activating system in the PMN development is underscored by the evidence that plasmin represents one of the most potent and versatile proteolytic enzymes capable of degrading a plethora of ECM proteins such as fibrin, fibronectin, vitronectin, and laminin, while also activating latent growth factors and cytokines and other proteases such as MMPs. This system could orchestrate the cascade of events, including ECM reprogramming, vascular remodeling, and the recruitment of pro-tumor immune cells into the niche. Furthermore, the plasminogen activation system operates through specific cell surface receptors like annexin A2/S100A10 complexes that can be delivered to distant organs via tumor-derived exosomes (or activated locally in the resident fibroblasts and immune cells). This provides a mechanistic pathway for primary tumors to condition future metastatic sites before cancer cell arrival remotely.

### 4.3. Cysteine Proteases and Cathepsin-Mediated PMN Modulation

Cysteine cathepsin proteases, particularly cathepsins B, L, and S, play complex roles in PMN formation by acting on both tumor and immune cells within the premetastatic microenvironment [89,159]. Cathepsin C (CTSC) has been identified as a critical factor secreted by lung-tropic breast cancer cells that promotes neutrophil recruitment and NET formation at premetastatic sites. MDSCs, key components of the PMN, express abundant cathepsin activity that contributes to their immunosuppressive functions and ability to differentiate into osteoclasts in bone metastatic sites. Interestingly, cathepsin activity appears to be downregulated during MDSC-osteoclast differentiation, suggesting that cathepsin inhibition may paradoxically promote osteoclastogenesis, a process implicated in bone metastasis [159]. This complexity underscores the importance of carefully considering timing and cell-specific targeting when developing cathepsin-based therapeutic interventions.

## 5. Organ-Specificity of Proteases in Premetastatic Niche Formation

Organ-specific protease activity plays a critical role in shaping the premetastatic niche (PMN) in different target organs. This organotropism is possible due to the unique ECM composition, local cell populations, and accessibility of vascular or stromal compartments [3,84]. The following analysis summarizes protease-dependent mechanisms in the PMN of the lung, liver, bone, and brain.

### 5.1. Lung PMN

The upregulation of multiple MMPs plays an important role in lung PMN formation. These MMPs include MMP2, MMP3, MMP9, MMP10, and MMP14. These MMPs are secreted by tumor cells, fibroblasts, neutrophils, and MDSCs [101,160]. MMP3 and MMP10 act synergistically with angiopoietin-2 to disrupt lung vasculature. This has been shown to increase permeability and facilitate the extravasation of circulating tumor cells [86]. Furthermore, myeloid cell–derived MMP9 has been shown to play a significant contribution to the lung PMN by remodeling the ECM, releasing growth factors, and recruiting additional immune cells [101,160].

Neutrophil elastase and other neutrophil proteases participate in the ECM breakdown. They also create NETs (neutrophil extracellular traps) and influence the recruitment and retention of both tumor and host cells in the developing PMN [160]. Furthermore, increased elastase activity correlated with greater intravascular arrest of metastatic cells in the lung.

The plasminogen/plasmin system also plays a significant role in lung PMN formation and is involved in both direct ECM degradation and the indirect activation of MMPs. Plasminogen binds to specific cell-surface receptors (including annexin II-S100A10, α-enolase, and uPAR) on endothelial cells, tumor cells, and immune cells within the lung microenvironment [123]. Upon activation by urokinase plasminogen activator (uPA) or tissue plasminogen activator (tPA), plasmin directly degrades fibronectin, laminin, and other ECM components. In addition, plasmin activates the inactive pro-MMPs to their active forms. This proteolytic cascade increases vascular permeability, a hallmark of lung PMN formation, and creates a permissive environment for the seeding of circulating tumor cells.

In conclusion, the redundant and cooperative actions of neutrophil elastase, cathepsin G, and multiple MMPs uniquely establish the lung PMN as a protease-rich environment. Lung-specific targets, such as Tsp-1, are degraded primarily in this organ rather than in other PMN sites. Furthermore, lung PMN formation is strongly potentiated by infections, smoking, or other lung-damaging stimuli, and these increase local neutrophil recruitment and protease release.

### 5.2. Liver PMN

Recent evidence, as discussed in the previous sections, has demonstrated that TIMP1 is the primary driver of hepatic PMN formation [88,109,161]. Moreover, TIMP1 induces expression of PMN markers in the liver, including CXCL12, fibronectin (FN1), TGFβ1, uPA, and S100A8. Increased TIMP1 levels enhance liver metastasis by upregulating uPA and activating Hepatic Growth Factor (HGF) signaling [162]. It will be interesting to evaluate whether hepatic PMN formation directly engages the other components of the plasminogen/plasmin activation system, specifically the identity of plasminogen receptors involved in modulating hepatic PMN. It is well established that the plasmin/plasminogen activation system supports HGF signaling in liver injury, ECM degradation, and enhanced motility of metastatic cells [163,164,165]. uPA also supports neutrophil-mediated release of pro-metastatic factors during niche conditioning [162]. Plasmin generation at the liver PMN is initiated by hepatic stellate cells and Kupffer cells. These cells express plasminogen receptors, resulting in localized proteolysis that remodels the hepatic ECM and promotes fibronectin deposition [166,167]. In the hepatic PMN, the uPA/uPAR/plasmin axis therefore sets the stage for metastatic colonization by modulating fibrin and ECM protein turnover to create “soft spots” in the liver stroma. This axis also activates pro-HGF to stimulate the invasive growth of both tumor cells and supportive hepatic parenchymal cells and promotes angiogenesis and immune cell trafficking via localized proteolytic gradients [162,163,168,169,170,171]. This specialized serine protease activity is a hallmark of liver-specific PMN biology and is a promising therapeutic target for disrupting organotropic metastasis.

Kupffer cells (liver-resident macrophages) and recruited immune cells secrete MMP9 and MMP14, enabling matrix remodeling and facilitating tumor cell invasion or colonization. In addition, primary tumor-derived signals upregulate MMP9 and MMP2 in hepatic sinusoidal lining cells. The classic MMP inhibitor, TIMP-1, can promote hepatic PMN recruitment acting indirectly, with CXCL12 (also known as CXCL12α) to recruit pro-tumorigenic neutrophils and drive collagen remodeling [88].

TIMP-1, elevated from primary tumors, induces liver-specific premetastatic niche formation through a multi-step cascade centered on neutrophil recruitment. Systemically elevated TIMP-1 triggers upregulation of CXCL12 specifically within hepatic tissue—a liver-specific response that is not observed in other organs. This hepatic CXCL12 gradient recruits CXCR4-expressing neutrophils from the circulation into the liver sinusoids via classic chemokine signaling; blockade of the CXCL12/CXCR4 axis with CXCR4 antagonists or antibodies prevents this recruitment and abrogates PMN formation and metastatic homing. Additionally, TIMP-1 signals through CD63 receptors on bone marrow myeloid progenitors to enhance granulopoiesis and systemic neutrophilia, expanding the pool of neutrophils available for hepatic recruitment. Once recruited to the liver, these neutrophils release an arsenal of proteases (uPA, MMPs, neutrophil elastase), activate hepatocyte growth factor (HGF) through uPA-mediated cleavage of pro-HGF, and upregulate pro-metastatic factors including fibronectin, TGFβ, and S100A8, thereby conditioning the hepatic microenvironment to support metastatic colonization. This TIMP-1/CXCL12/CXCR4/neutrophil axis is uniquely operative in the liver, reflecting the hepatic stellate cell response, fenestrated sinusoidal architecture, and fibrosis-like remodeling pathways [88,162,172,173,174]. This makes it a liver-specific mechanism distinct from TIMP-1 biology in other organs and a potential therapeutic target through CXCR4 antagonism, anti-TIMP-1 strategies, or also by transient neutrophil depletion. Clinically, elevated circulating TIMP-1 levels in cancer patients may serve as a biomarker for hepatic PMN formation and liver metastasis risk. This allows stratification of high-risk patients for adjuvant CXCR4 antagonist therapy (e.g., plerixafor) in the perioperative window before overt metastases develop. Preclinical studies using pancreatic, colorectal, and breast cancer models has shown that CXCR4 blockade, anti-CD63 neutralization, or transient anti-Ly6G neutrophil depletion can effectively disrupt TIMP-1-induced liver PMN formation and prevent hepatic metastatic seeding [88,174]. This supports clinical trials combining primary tumor resection with TIMP-1-targeting or neutrophil-modulating therapies in patients at high risk for liver metastasis [175].

### 5.3. Bone PMN

Bone is distinct in its reliance on cysteine cathepsins (notably cathepsins B, L, and X), which regulate the activity of osteoclasts and the remodeling of the bone matrix. Cathepsin-driven collagen and bone ECM proteolysis sets the stage for tumor cell infiltration, and bone metastatic tumor cells elevate local MMP expression, increasing presentation or activity of RANKL (Receptor Activator of Nuclear factor Kappa-Β Ligand), thereby promoting osteoclast differentiation and bone breakdown [176,177,178,179].

Plasmin contributes to bone metastasis by degrading type I collagen, a key component of the bone matrix. Breast cancer cells degrade bone collagen, and this requires both plasminogen activation and MMP activity, illustrating the cooperative nature of proteolytic mechanisms [123,180,181]. The uPA/uPAR/plasmin system at the bone PMN facilitates osteoclast-independent and osteoclast-dependent ECM degradation and enhances metastatic cell accessibility. The uPA/uPAR activity by the tumor cells arriving at the bone not only degrades the bone ECM, such as collagen I, fibronectin, and laminin through plasmin-mediated proteolysis, but also simultaneously promotes osteoclast differentiation and activity [168,182].

### 5.4. Brain PMN

Brain tropism is strongly associated with cathepsin S, a cysteine protease produced by both tumor cells and macrophages [14]. Cathepsin S cleaves junctional adhesion molecule B (JAM-B) at the blood–brain barrier (BBB), facilitating tumor cell transmigration specifically into the brain parenchyma [90].

Brain stromal cells, particularly astrocytes, secrete plasminogen activators (tPA, uPA) that activate plasmin, which usually impairs tumor survival via FasL-mediated apoptosis. Brain-metastatic cancer cells often express serpins (protease inhibitors) such as neuroserpin and serpin B2 to counteract this otherwise anti-metastatic protease-rich microenvironment. This suggests that in the brain PMN, the balance between plasminogen activation and serpin-mediated inhibition is a critical determinant of metastatic success [90,183,184].

Therefore, cathepsins (especially B, K, and L) are much more dominant in bone than in lung or liver PMN as reflected in the bone collagen-rich matrix and reliance on osteoclast-driven resorption for PMN conditioning. Bone is the only PMN in which cathepsin K is a key prometastatic protease due to its osteoclast specificity. Bone PMN formation relies less on serine protease pathways and more on cysteine cathepsin and MMP interplay for both ECM degradation and osteoclast activation [177,185,186].

## 6. Proteases as Therapeutic Targets in Premetastatic Niche Formation

As discussed, the PMN is initiated by the systemic secretion of factors that both directly activate changes in organs and also cause the activation of tissue-resident cells and the recruitment of cells to these organs. Thus, these distant organs are prepared to receive circulating tumor cells, thereby creating an environment conducive to metastatic colonization. PMN formation is dependent on the coordinated action of multiple protease families that participate in extracellular matrix remodeling, immune modulation, angiogenesis, and inflammatory signaling. Proteases have emerged as important therapeutic targets in the formation and development of PMNs. The targeting of proteases that participate in PMN formation represents a fundamental shift in cancer treatment strategies, moving from targeting established metastases, which have been largely unsuccessful in preventing deaths from metastases, to preventing their formation altogether [7,187,188].

Despite their central role in PMN formation, early clinical trials with broad-spectrum MMP inhibitors were unsuccessful due to their application in advanced-stage patients and the occurrence of significant adverse effects, leading to the development of more selective inhibitors with improved pharmacokinetic properties [189,190,191].

Current therapeutic approaches targeting proteases in PMN formation involve multiple strategies, including direct enzyme inhibition, modulation of protease expression, and targeting of protease-activated pathways. TIMP-1 has been shown to create hepatic premetastatic niches through the upregulation of CXCL12, fibronectin, and TGF-β1, as well as the recruitment of neutrophils. CXCR4 antagonists, such as AMD3100, have demonstrated efficacy in reducing TIMP-1-induced niche formation [88]. Low-dose epigenetic therapy, combining azacytidine and entinostat, has shown promise in disrupting PMN formation by modulating the recruitment and activity of myeloid-derived suppressor cells [192]. While biomarkers such as hepatocyte growth factor, E-selectin, and IL-6 have been identified as necessary in PMN formation, clinical trials specifically targeting PMN endpoints remain limited. The development of more selective protease inhibitors, including antibody-based therapeutics and structure-based inhibitors, offers improved specificity and reduced toxicity compared to earlier broad-spectrum approaches.

The therapeutic targeting of proteases in PMN formation represents a paradigm shift toward preventing metastasis rather than treating established disease. Success in this approach requires an understanding of protease biology, improved patient selection strategies, and careful timing of interventions to disrupt PMN formation without disrupting essential physiological processes. As our knowledge of protease-mediated PMN mechanisms continues to evolve, these therapeutic strategies hold significant promise for improving cancer patient outcomes by preventing the metastatic cascade at its earliest stages. The goal of this approach is to change metastatic cancer from a lethal disease to a chronic disease.

Targeting protease-mediated PMN formation could prevent metastasis rather than merely treat established lesions [3,84]. Three approaches have the potential to target proteases in PMN. First, the success of adjuvant epigenetic therapy in disrupting MDSC-derived proteases and preventing experimental lung metastases [192] has validated a specific approach. Second, the identification of cathepsin S as a therapeutic target for brain metastasis prevention [90] has identified a druggable target for brain PMN. Third, elucidation of MMP-mediated vascular destabilization in premetastatic lungs [86] has reinvigorated efforts to develop effective MMP inhibitors. Future clinical trials will therefore utilize appropriate timing (in the adjuvant setting), biomarker-guided patient selection, organ-specific targeting, and rational combinations to fully optimize PMN-targeted therapy and its role in preventing cancer metastasis.

### 6.1. Adjuvant Epigenetic Therapy

The most compelling clinical evidence for PMN-targeted therapy comes from adjuvant low-dose epigenetic therapy (LD-AET) using azacytidine and entinostat. Lu et al. [192] demonstrated that after surgical removal of primary lung, breast, and esophageal cancers, the administration of LD-AET disrupted the premetastatic microenvironment. This resulted in significant inhibition of lung metastases by selectively targeting MDSCs. This approach works through two complementary mechanisms: (1) inhibiting MDSC trafficking to the premetastatic lung via downregulation of CCR2 (monocytic MDSCs) and CXCR2 (polymorphonuclear MDSCs); (2) promoting MDSC differentiation to a more interstitial macrophage-like phenotype that no longer supports PMN formation [192]. Interestingly, LD-AET was shown to reduce protease expression in the PMN microenvironment by decreasing MDSC levels. MDSCs are significant sources of MMP-9, neutrophil elastase, and other proteases that remodel the premetastatic ECM [3,192]. By depleting MDSCs from premetastatic lungs, LD-AET indirectly suppresses the proteolytic cascade essential for PMN maturation. Interestingly, mice treated with LD-AET showed decreased MDSC accumulation in premetastatic lungs, longer disease-free survival, and increased overall survival compared to chemotherapy [192]. A small phase II clinical trial (13 patients with stage I NSCLC) was initiated but terminated prematurely due to logistical issues, though preliminary data suggested improved outcomes [192,193,194].

This work has identified three critical principles for PMN-targeted therapy: (1) timing is absolutely critical and intervention must occur in the adjuvant setting after primary tumor resection, during active PMN formation; (2) indirect protease inhibition via immune cell modulation can be as effective as direct enzyme inhibition; (3) a therapeutic window exists between primary tumor removal and metastatic seeding [192,193].

### 6.2. Organ-Specific Protease Inhibition Strategies

Huang et al. found that angiopoietin-2, MMP-3, and MMP-10 are synergistically upregulated in premetastatic lungs. These elements disrupt vascular integrity and increase permeability—a prerequisite for the extravasation of circulating tumor cells. Lentivirus-based RNA interference targeting these three factors attenuated pulmonary vascular permeability in vivo, suppressed myeloid cell infiltration, and significantly inhibited spontaneous lung metastasis in orthotopic breast cancer models [86]. This work demonstrated that MMP inhibition specifically in the premetastatic phase—rather than in established tumors—can effectively prevent metastasis.

Modern approaches to targeting lung PMNs include nanoparticle delivery systems that exploit PMN-specific features. It has been reported that pH-responsive pHLIP (pH Low Insertion Peptide) accumulates specifically in acidic premetastatic lungs. When conjugated to dexamethasone, the compound effectively attenuates lung metastatic burden by disrupting pro-inflammatory responses [195]. Similarly, self-delivery nanoparticles containing low-molecular-weight heparin prevented lung PMN formation by blocking P-selectin/PSGL-1-mediated MDSC recruitment and also suppressed MMP-9 expression, thereby inhibiting circulating tumor cell colonization [196].

Sevenich et al. identified cathepsin S as a critical mediator of breast-to-brain metastasis through comprehensive proteomic analysis of tumor- and stroma-supplied proteolytic networks in different metastatic microenvironments. Cathepsin S, produced by both macrophages and tumor cells, cleaves junctional adhesion molecule B (JAM-B) at the blood–brain barrier. This facilitates tumor cell transmigration specifically into the brain parenchyma [90]. High cathepsin S expression at the primary tumor site correlates with decreased brain metastasis-free survival in breast cancer patients. This suggests a potential biomarker for patient stratification. Combined depletion of cathepsin S in both tumor cells and stromal macrophages can significantly reduce the incidence of brain metastasis. However, targeting either source separately had no effect [90]. This finding underscores the redundancy of protease cellular origins in PMN formation and the need for comprehensive protease-inhibition strategies.

Pharmacological inhibition of cathepsin S using the selective inhibitor VBY-999 significantly reduced experimental brain metastasis when administered before tumor seeding and throughout progression. This suggested that cathepsin S is an important therapeutic target for preventing brain metastases [90,197].

The hepatic PMN presents a therapeutic paradox: elevated tissue inhibitor of metalloproteinase-1 (TIMP-1) promotes rather than prevents liver metastasis. Seubert et al. demonstrated that TIMP-1 promotes hepatic PMN by a mechanism involving CXCL12α-mediated neutrophil recruitment and host-derived urokinase plasminogen activator (uPA), which mediates TIMP-1-triggered hepatocyte growth factor signaling and tumor cell dissemination throughout the liver parenchyma [88]. This suggests that blocking TIMP-1 or interfering with the TIMP-1/uPA axis may prevent liver-specific metastasis. This approach, therefore, highlights the organ-specific and context-dependent nature of protease function in PMN formation.

In conclusion, each organ displays a unique protease signature reflecting its tissue-specific architecture, ECM composition, and resident cell populations. Lung relies on neutrophil serine proteases degrading Tsp-1; liver uniquely uses TIMP-1/neutrophil/uPA axis; bone depends on osteoclast cathepsin K; brain requires cathepsin S-mediated BBB disruption. These distinct mechanisms enable organ-specific therapeutic targeting.

### 6.3. Technical Challenges in Real-Time PMN Dynamics Analysis Using In Vivo Imaging

Real-time analysis of premetastatic niche (PMN) dynamics using in vivo imaging faces major technical barriers. These barriers involve detection, temporal dynamics, spatial resolution, and biological validation. PMN formation occurs at microscopic sites and lacks clear anatomical boundaries. This makes reliable identification and distinction from normal physiology very difficult. Early PMNs form before overt metastases are detectable and are therefore difficult to target for imaging prospectively. PMN formation occurs over days to weeks with both unpredictable timing and anatomical location, thereby requiring prolonged longitudinal monitoring with implantable chronic imaging windows that must maintain biocompatibility over extended periods without inducing local or systemic inflammation that confounds biological readouts. Multi-scale imaging requirements—from subcellular resolution (~0.25 μm) for measuring proteolytic activity to entire organ volumes (~1–2 mm^3^)—demand computationally intensive, large-volume high-resolution (LVHR) imaging with hundreds of image mosaics and sophisticated stitching algorithms. However, organ-specific architecture creates distinct barriers. For example, lung respiratory motion and fenestrated vasculature complicate stable imaging, and the liver sinusoidal network and rapid blood flow require specialized surgical windows.

Furthermore, the brain skull creates optical barriers that compromise resolution, and the bone’s mineralized matrix is optically impenetrable. Dynamic cellular and molecular measurements face the challenge of distinguishing overlapping myeloid cell populations (neutrophils, MDSCs, macrophages) with similar morphology and phenotypic plasticity, while real-time assessment of functional parameters, including protease activity, pH gradients, oxygen tension, and metabolic states, requires non-invasive biosensors that are technically demanding to multiplex and interpret. Validation of PMN biology is confounded by artifactual perturbations caused by surgical window implantation, anesthesia, and animal handling, which alter immune cell dynamics. Clinical translation is fundamentally limited by the inability to prospectively identify and image premetastatic organs in asymptomatic human patients, due to ethical and technical constraints. Yet, despite these barriers, emerging solutions—including improved implantable chronic imaging windows, microcartography techniques for relocalization, engineered biomaterial PMN models with defined anatomy, and advanced biosensors for real-time functional measurement—are incrementally reducing technical impediments and enabling more sophisticated longitudinal tracking of PMN formation and protease-mediated conditioning [3,198,199,200]. As Entenberg et al. (2023) demonstrate, surgical engineering approaches are lowering barriers to intravital imaging of cancer progression and metastasis [199], while Aguado et al. (2017) show that engineered PMNs provide platforms for studying mechanisms in controlled settings [200]. The integration of these emerging technologies with the current understanding of organ-specific PMN biology promises to accelerate the discovery of critical protease-driven conditioning mechanisms (reviewed [3,17,83]).

### 6.4. Critical Success Factors for Future PMN-Targeted Protease Therapies

Based on the successes and failures of protease-targeted therapies, several principles emerge for effective PMN disruption [3,84,192]. These include three important principles. First, therapeutic timing as an intervention must occur in the adjuvant or neoadjuvant setting in high-risk patients after primary tumor resection, during the window when PMNs are actively forming but metastases have not yet established. This contrasts with failed MMP inhibitor trials that enrolled patients with advanced metastatic disease, long after the critical PMN formation phase had passed [86,192]. Secondly, biomarker-guided patient selection is also essential because not all patients develop robust PMNs. Circulating levels of PMN-promoting factors (S100A8/A9, tumor-derived exosomes), protease expression profiles at the primary site (cathepsin S for brain tropism), or imaging biomarkers identifying premetastatic organ changes should guide patient enrollment in these trials [3,90]. Third, organ-specific targeting is an obvious principle. For example, lung PMNs rely on MMP-3/MMP-10/angiopoietin-2 [86,100], brain PMNs on cathepsin S [90], liver PMNs on TIMP-1/uPA [88], and bone PMNs on cathepsins and MMPs. Therapies must account for these organ-specific protease networks rather than applying broad-spectrum inhibition [84].

In addition to these principles, there are other important considerations. These include combination approaches due to redundancy in PMN formation pathways. Therefore, combination strategies targeting multiple protease families or coupling protease inhibition with immune modulation (as demonstrated by [192]) will likely be more effective than monotherapy. Furthermore, rather than directly inhibiting proteases, depleting or reprogramming the cellular sources (MDSCs, TAMs, neutrophils) that produce PMN-conditioning proteases may be more clinically effective, as demonstrated by the LD-AET approach [192,193]. Key to targeting the PMN will be effective delivery systems. For example, nanoparticle-mediated delivery and pH-responsive targeting that selectively accumulate in premetastatic organs might enhance therapeutic responses with minimum toxicity [195,196,201].

Therefore, the success of disrupting MDSC-derived proteases and preventing experimental lung metastases [192], the identification of cathepsin S as a therapeutic target for brain metastasis prevention [90], and the elucidation of MMP-mediated vascular destabilization in premetastatic lungs [86] have provided key clues to targeting the PMN. Future clinical trials will therefore utilize the principles of appropriate timing (in the adjuvant setting), biomarker-guided patient selection, organ-specific targeting, and rational combinations to allow successful targeting of the PMN and, therefore, prevent cancer metastasis.

## 7. Conclusions

The emerging understanding of extracellular protease roles in PMN formation represents a paradigm shift in cancer metastasis research, revealing how MMPs, serine proteases, and cysteine cathepsins orchestrate the systematic preparation of distant organs for metastatic colonization. Key findings demonstrate that proteases function through multiple interconnected mechanisms: MMPs such as MMP9 and MMP2 facilitate ECM remodeling and immune cell recruitment, while the plasminogen–plasmin system serves as an overall master regulator through its broad-spectrum proteolytic activity and ability to activate other proteases. The serine proteases, particularly neutrophil elastase and cathepsin G, create pro-metastatic microenvironments by degrading anti-angiogenic proteins. Meanwhile, S100A10-mediated plasmin generation provides a mechanistic pathway for the remote conditioning of metastatic sites. Understanding these protease-mediated mechanisms is crucial because they represent the earliest window for intervention in the metastatic cascade, offering the potential to transform metastatic cancer from a lethal disease into a chronic, manageable condition by preventing niche formation rather than treating established metastases.

However, significant methodological challenges limit current PMN research and highlight critical future directions for this field. Most studies rely predominantly on orthotopic and transgenic mouse models focusing on pulmonary PMN, which provide limited information about multi-organ metastasis due to the restricted availability of appropriate syngeneic models. The transient nature of PMN formation demands temporal precision in analysis, while distinguishing protease activity in normal tissues from PMN-specific activity requires sophisticated in vivo measurement techniques specific to each protease family. To address the fundamental question of how tumor-derived secreted factors (TDSFs) reach and condition distant PMN sites, future research must develop novel technologies, including organ-specific PMN biomarkers for clinical detection, advanced imaging techniques to track TDSF circulation and uptake, and improved methodologies to measure real-time protease activity in vivo. Additionally, a systematic investigation using plasminogen knockout mouse models and protease inhibitors, with spatio-temporal analysis across various PMN organs (brain, lungs, liver), will be essential to definitively establish the individual and collective contributions of different protease families in the complex, redundant proteolytic networks that drive PMN formation.

Most methods for studying PMN primarily rely on orthotopic and transgenic mouse models and predominantly focus on pulmonary PMN. Although highly relevant, such pre-clinical models can provide limited information on metastasis and are limited by the availability of appropriate syngeneic models to test multi-organ PMNs. Furthermore, the limited understanding and availability of biomarkers to detect the presence of PMN in patients necessitate the development of novel technologies to identify organ-specific markers of PMN in patients. Furthermore, several challenges exist in studying proteases during development. Firstly, we require temporal precision in analyzing the PMN as its formation can be transient. Secondly, whole-animal protease activity requires sophisticated in vivo measurements, specific to each protease. Thirdly, it is crucial to distinguish protease activity in normal tissue from that in the PMN. Finally, redundancy in protease activity makes it challenging to assess the individual contributions of each protease family.

## Figures and Tables

**Figure 1 biomolecules-15-01696-f001:**
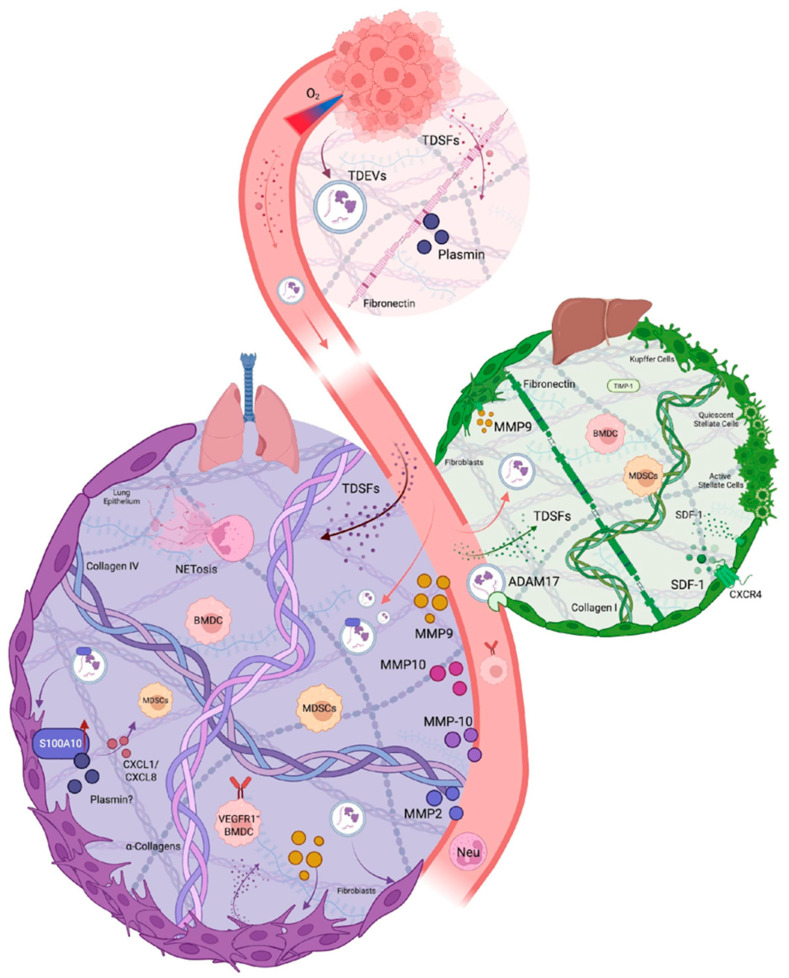
Tracing Proteases Active in the PMN. Hypoxic environments at the primary tumor promote the release of TDSFs and TDEVs into circulation. Altered vasculature allows molecules to enter the specific organ ECM, where proteases recruit immune cells, remodel the ECM, and form the future site of metastasis. The bulk of PMN research has focused on the lung and liver PMN due to technical constraints.

**Table 1 biomolecules-15-01696-t001:** Evidence map of proteases implicated in PMN formation.

Protease (Subfamily)	Primary Source at PMN	Substrate	Principal PMN Function	PMN Organs (Function)	Mouse Model	Reference
MMP-9	BMDCs/VEGFR1^+^ clusters	Collagen IV	ECM remodelling, vascular permeability, and immunosuppressive cell recruitment	Lungs	Orthotopic/Knock-out	[5,35]
MMP-3/ MMP-10	Lung stromal/endothelial cells	Basement membrane proteins; Proteoglycans, fibronectin, Laminin, Collagen	Vascular Permeability, ECM remodelling (basement membrane degradation)	Lungs	B16/F10 mouse tumor model, mouse RNAi injections	[86]
MMP2	Cd11b^+^ myeloid cells	Collagen	Recruitment of BMDCs and metastasizing tumor cells	Lungs	Orthotopic breast cancer models	[25]
LOX/LOXL	Tumor secreted/CAFs	Collagen crosslinking	ECM remodelling, collagen crosslinking, BMDC adhesion and invasion	Lungs (mouse) Multiorgan (human)	Orthotopic breast cancer model	[25]
Neutrophil elastase/Cathepsin G	Neutrophils	Tsp-1 degradation	Tsp-1 degradation (inhibitor of tumor angiogenesis and growth)	Lungs	Orthotopic B16-BL6 mouse tumor model/LPS inflammation, B16 and LLC experimental metastasis model, and Knock-out mouse models	[87]
TIMP-1	Tumor	MMPs	Enhances hepatic SDF-1 production, recruits neutrophils to the liver through the SDF-1/CXCR4 axis	Liver	Various Tumor models—syngeneic and xenograft	[88]
Cathepsin C	Tumor	NE, PR3, granzymes A and B	Neutrophil infiltration through PR3 and the inflammatory cascade, NET formation and TSP-1 degradation	Lungs	Various mammary tumor models, transgenic overexpression and knock-out models, spontaneous and experimental metastasis models	[89]
Cathepsin S	Macrophages, tumor cells	JAM-B at the blood–brain barrier	Blood–brain barrier disruption	Brain	Breast cancer brain metastasis models, cathepsin S KO	[90]
ADAM17	Tumor exosomes	Membrane-bound proteins: cytokines, growth factors, VE-Cadherin	Adherens junction destabilization and disruption of vascular permeability	Liver	In vitro permeability models, xenograft models, no true PMN mouse models	[91]
Plasminogen–Plasmin	Lung fibroblasts, hepatic stellate cells, neutrophils, endothelial cells	Fibrin, Fibronectin, Laminin, pro-MMPs (MMP2, MMP9), pro-HGF, Collagen	ECM remodeling, MMP activation, HGF activation, vascular permeability, MDSC recruitment (via S100A10-mediated chemokine expression), neutrophil trafficking	Lung, liver	S100A10 KO, uPA KO, B16/F10, E0771, orthotopic breast cancer/A	[88,92,93]

MMP—Matrix metalloproteases, LOX—Lysyl Oxidase, TIMP-1—Tissue Inhibitor of Metalloprotease-1, PR3—Proteinase 3, NET—Neutrophil Extracellular Traps, TSP-1 Thrombospondin 1, VE-cadherin—Vascular Endothelial cadherin.

## Data Availability

No new data were created or analyzed in this study.

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
