# Peer review of "The Role of Extracellular Proteases and Extracellular Matrix Remodeling in the Pre-Metastatic Niche"

_biomolecules, 2025, doi:10.3390/biom15121696_

Round 1
Reviewer 1 Report
Comments and Suggestions for Authors
The authors presented the review entitled “The Role of Extracellular Proteases and Extracellular Matrix Remodeling in the Pre-metastatic Niche”.
I suggest figures to show Pre-metastatic Niche, concepts and factors.
Author Response
We have added a figure and a graphical abstract as per your request
Reviewer 2 Report
Comments and Suggestions for Authors
The manuscript analyzes the relevance of "extracellular proteases and extracellular matrix remodeling in the pre-metastatic niche" (PMN).
The authors listed the enzymes involved in remodeling the environment to favor the engraftment of metastatic cells.
The review is on a very hot and interesting topic. The authors did not add any figures to explain and summarize the huge amount of information given. This presentation does not help the reader to follow the content of the manuscript.
In addition, the authors stated that some tumor-derived secreted factors (TDSF) are responsible for the generation of the pre-metastatic niche. Unfortunately, the authors did not clearly list these factors. On the contrary, the manuscript https://doi.org/10.1038/s41392-024-01937-7 lists these factors. Without knowing these factors, it is almost impossible to understand the content of the manuscript.
The role of the immune system is not analyzed in detail like in the paper cited, nor is the role of cellular components in the generation of PMN.
Overall, the manuscript is not comprehensive but limited, giving a general and not detailed scenario of the topic.
Author Response
We apologize to the author if we did not make it clear that our intention was to only provide a detailed review of the proteases of the PMN. To provide a detailed review of the PMN in general would be out of the scope of the review and has been accomplished by several other authors--what had not been done before was a detailed review of the proteases of the PMN--our goal. Secreted factors such as S100A9/9, Lox, etc. are discussed in the text, but since the link between these factors and the PMN proteases is only poorly understood, we focused our efforts on defining the key proteases of the PMN.
However, we do quote the reference that you refer to--we have over 200 references cited.--
"The role of the immune system is not analyzed in detail like in the paper cited, nor is the role of cellular components in the generation of PMN."--absolutely, as discussed, we have focused on the proteases and not the immune cells, as that could be a separate review by itself.
We have added a figure and a graphical abstract as per your request.
Reviewer 3 Report
Comments and Suggestions for Authors
The premetastatic niche (PMN) is a specialized microenvironment formed in distant organs before cancer cells arrive, representing a reconceptualization of metastasis as an active, stepwise process. Extracellular proteases such as MMPs, serine proteases, and cathepsins induce extracellular matrix remodeling, modulate immune responses, and alter vascular permeability, thereby promoting metastasis site formation. Specifically, MMP9/2, TIMP-1, the plasminogen system, and neutrophil elastase have been reported as major factors in PMN formation. While therapeutically suppressing PMN formation is gaining attention, challenges remain regarding organ-specific biomarkers and methods for evaluating protease activity in vivo.
This research is highly intriguing, but several concerns warrant discussion.
- What organ-specificity exists in the role of proteases in PMN formation across different organs?
- What is the molecular mechanism by which TIMP-1 induces liver-specific PMNs via neutrophils?
- What technical challenges exist in real-time PMN dynamics analysis using in vivo imaging?
- Please discuss clinical applications.
Good.
Author Response
Thank you for your positive comments. As per your request, we have thoroughly checked the grammar and made multiple changes. We use Grammerly for these edits so we are confident that all grammatical weaknesses have been corrected.
In order to answer your 4 questions we have added a new chapter--5. This is directed to "What organ-specificity exists in the role of proteases in PMN formation across different organs"
To answer the question about TIMP and liver we have extended the paragraph and added section 5.2 --"In contrast, tissue inhibitor of metalloproteinases (TIMP)-1 is critical for liver metastasis in colorectal cancer models. TIMP-1 is a small molecular weight secreted protein initially identified as an endogenous inhibitor of MMPs [97]."
For technical challenges--we clarrified--6.3 Technical Challenges in Real-Time PMN Dynamics Analysis Using In Vivo Imaging
Real-time analysis of premetastatic niche (PMN) dynamics using in vivo imaging faces major technical barriers. These barriers involve detection, temporal dynamics, spatial resolution...
Clinical applications are discussed in 6.4--for example-6.3 Technical Challenges in Real-Time PMN Dynamics Analysis Using In Vivo Imaging--Real-time analysis of premetastatic niche (PMN) dynamics using in vivo imaging faces major technical barriers. These barriers involve detection, temporal dynamics, spatial resolution...
The response to your comments has substantially strengthened the manuscript--thank you.
Reviewer 4 Report
Comments and Suggestions for Authors
In their comprehensive review, Okura and colleagues present up‑to‑date information on the proteolytic molecules and pathways involved in the formation of the premetastatic niche, which enables tumor cells from a distant site to enter and survive in a new microenvironment. The focus is on matrix metalloproteinases (MMPs), but other extracellular proteases, such as serine proteases and cysteine cathepsins, are also discussed. This knowledge is expected to contribute to preventing metastasis formation in clinical settings. A well‑balanced selection of animal models and experimental metastasis assays is presented, providing a better understanding of this essential and still highly relevant topic in cancer research.
This review is highly informative for physicians, cancer researchers, and students interested in this field. I enjoyed reading it.
Although a schematic representation of the metastatic niche (and the metastatic process) summarizing the major molecules discussed could have been useful, it is not absolutely necessary.
Please check the nomenclature to ensure whether “SDF‑1” or “CXCL12” is more appropriate, and note somewhere that both refer to the same molecule. Since other chemokines are cited using modern nomenclature, I suggest using “CXCL12.”
Since it may escape the typing editor: „remetastatic“ (p is missing: premetastastic)
Author Response
Thank you for your positive comments--we have added a figure and a graphical abstract. Also, “SDF‑1” or “CXCL12"---we have changed the manuscript to CXCL12. The addition of section 5 that looks at specific proteases in different niches (liver, brain, bone, lung) should also help with the clarity.
Round 2
Reviewer 2 Report
Comments and Suggestions for Authors
The authors replied to the reviewer's concerns. They added a lot of information in response to the other reviewer's queries on the role of proteases in PMN in different organs.
Reviewer 3 Report
Comments and Suggestions for Authors
The authors replied well, so the manucript is suitable for publication.
Comments on the Quality of English LanguageGood.